# Online Learning in Markov Decision Processes with Adversarially Chosen Transition Probability Distributions

**Yasin Abbasi-Yadkori**
Queensland University of Technology
yasin.abbasiyadkori@qut.edu.au

**Peter L. Bartlett**
UC Berkeley and QUT
bartlett@eecs.berkeley.edu

**Varun Kanade**
UC Berkeley
vkanade@eecs.berkeley.edu

**Yevgeny Seldin**
Queensland University of Technology
yevgeny.seldin@gmail.com

**Csaba Szepesvári**
University of Alberta
szepesva@cs.ualberta.ca

## Abstract

We study the problem of online learning Markov Decision Processes (MDPs) when both the transition distributions and loss functions are chosen by an adversary. We present an algorithm that, under a mixing assumption, achieves $O(\sqrt{T \log |\Pi|} + \log |\Pi|)$ regret with respect to a comparison set of policies $\Pi$. The regret is independent of the size of the state and action spaces. When expectations over sample paths can be computed efficiently and the comparison set $\Pi$ has polynomial size, this algorithm is efficient.

We also consider the episodic adversarial online shortest path problem. Here, in each episode an adversary may choose a weighted directed acyclic graph with an identified start and finish node. The goal of the learning algorithm is to choose a path that minimizes the loss while traversing from the start to finish node. At the end of each episode the loss function (given by weights on the edges) is revealed to the learning algorithm. The goal is to minimize regret with respect to a fixed policy for selecting paths. This problem is a special case of the online MDP problem. It was shown that for randomly chosen graphs and adversarial losses, the problem can be efficiently solved. We show that it also can be efficiently solved for adversarial graphs and randomly chosen losses. When both graphs and losses are adversarially chosen, we show that designing efficient algorithms for the adversarial online shortest path problem (and hence for the adversarial MDP problem) is as hard as learning parity with noise, a notoriously difficult problem that has been used to design efficient cryptographic schemes. Finally, we present an efficient algorithm whose regret scales linearly with the number of distinct graphs.

## 1  Introduction

In many sequential decision problems, the transition dynamics can change with time. For example, in steering a vehicle, the state of the vehicle is determined by the actions taken by the driver, but also by external factors, such as terrain and weather conditions. As another example, the state of a

robot that moves in a room is determined both by its actions and by how people in the room interact with it. The robot might not have influence over these external factors, or it might be very difficult to model them. Other examples occur in portfolio optimization, clinical trials, and two player games such as poker.

We consider the problem of online learning Markov Decision Processes (MDPs) when the transition probability distributions and loss functions are chosen adversarially and are allowed to change with time. We study the following game between a learner and an adversary:

1. The (oblivious) adversary chooses a sequence of transition kernels $m_t$ and loss functions $\ell_t$.

2. At time $t$:
    (a) The learner observes the state $x_t$ in state space $\mathcal{X}$ and chooses an action $a_t$ in the action space $\mathcal{A}$.
    (b) The new state $x_{t+1} \in \mathcal{X}$ is drawn at random according to distribution $m_t(\cdot|x_t, a_t)$.
    (c) The learner observes the transition kernel $m_t$ and the loss function $\ell_t$, and suffers the loss $\ell_t(x_t, a_t)$.

To handle the case when the representation of $m_t$ or $\ell_t$ is very large, we assume that the learner has a black-box access to $m_t$ and $\ell_t$. The above game is played for a total of $T$ rounds and the total loss suffered by the learner is $\sum_{t=1}^{T} \ell_t(x_t, a_t)$. In the absence of state variables, the MDP problem reduces to a *full information online learning problem* (Cesa-Bianchi and Lugosi [1]). The difficulty with MDP problems is that, unlike full information online learning problems, the choice of a policy at each round changes the future states and losses.

A policy is a mapping $\pi : \mathcal{X} \to \Delta_{\mathcal{A}}$, where $\Delta_{\mathcal{A}}$ denotes the set of distributions over $\mathcal{A}$. To evaluate the learner's performance, we imagine a hypothetical game where at each round the action played is chosen according to a fixed policy $\pi$, and the transition kernels $m_t$ and loss functions $\ell_t$ are the same as those chosen by the oblivious adversary. Let $(x_t^\pi, a_t^\pi)$ denote a sequence of state and action pairs in this game. Then the loss of the policy $\pi$ is $\sum_{t=1}^{T} \ell_t(x_t^\pi, a_t^\pi)$. Define a set $\Pi$ of policies that will be used as a benchmark to evaluate the learner's performance. The regret of a learner $A$ with respect to a policy $\pi \in \Pi$ is defined as the random variable $R_T(A, \pi) = \sum_{t=1}^{T} \ell_t(x_t, a_t) - \sum_{t=1}^{T} \ell_t(x_t^\pi, a_t^\pi)$. The goal in adversarial online learning is to design learning algorithms for which the regret with respect to any policy grows sublinearly with $T$, the total number of rounds played. Algorithms with such a guarantee, somewhat unfortunately, are typically termed *no-regret* algorithms.

We also study a special case of this problem: the episodic online adversarial shortest path problem. Here, in each episode the adversary chooses a layered directed acyclic graph with a unique start and finish node. The adversary also chooses a loss function, i.e., a weight for every edge in the graph. The goal of the learning algorithm is to choose a path from start to finish that minimizes the total loss. The loss along any path is simply the sum of the weights on the edges. At the end of the round the graph and the loss function are revealed to the learner. The goal, as in the case of the online MDP problem, is to minimize regret with respect to a class of policies for choosing the path. Note that the online shortest path problem is a special case of the online MDP problem; the states are the nodes in the graph and the transition dynamics is specified by the edges.

## 1.1 Related Work

Burnetas and Katehakis [2], Jaksch et al. [3], and Bartlett and Tewari [4] propose efficient algorithms for finite MDP problems with stochastic transitions and loss functions. These results are extended to MDPs with large state and action spaces in [5, 6, 7]. Abbasi-Yadkori and Szepesvári [5] and Abbasi-Yadkori [6] derive algorithms with $O(\sqrt{T})$ regret for linearly parameterized MDP problems, while Ortner and Ryabko [7] derive $O(T^{(2d+1)/(2d+2)})$ regret bounds under a Lipschitz assumption, where $d$ is the dimensionality of the state space. We note that these algorithms are computationally expensive.

Even-Dar et al. [8] consider the problem of online learning MDPs with fixed and known dynamics, but adversarially changing loss functions. They show that when the transition kernel satisfies a mixing condition (see Section 3), there is an algorithm with regret bound $O(\sqrt{T})$. Yu and Mannor [9, 10] study a harder setting, where the transition dynamics may also change adversarially over time.

However, their regret bound scales with the amount of variation in the transition kernels and in the worst case may grow linearly with time.

Recently, Neu et al. [11] give a no-regret algorithm for the episodic shortest path problem with adversarial losses but stochastic transition dynamics.

## 1.2 Our Contributions

First, we study a general MDP problem with large (possibly continuous) state and action spaces and adversarially changing dynamics and loss functions. We present an algorithm that guarantees $O(\sqrt{T})$ regret with respect to a suitably small (totally bounded) class of policies $\Pi$ for this online MDP problem. The regret grows with the metric entropy of $\Pi$, so that if the comparison class is the set of all policies (that is, the algorithm must compete with the optimal fixed policy), it scales polynomially with the size of the state and action spaces. The above algorithm is efficient as long as the comparison class has polynomial size and we can compute expectations over sample paths for each policy. This result has several advantages over the results of [5, 6, 7]. First, the transition distributions and loss functions are chosen adversarially. Second, by designing an appropriate small class of comparison policies, the algorithm is efficient, even in the face of very large state and action spaces.

Next, we present efficient no-regret algorithms for the episodic online shortest path problem for two cases: when the graphs and loss functions (edge weights) are chosen adversarially and the set of graphs is small; and when the graphs are chosen adversarially, but the loss is stochastic.

Finally, we show that for the general adversarial online shortest path problem, designing an efficient no-regret algorithm is at least as hard as learning parity with noise. Since the online shortest path problem is a special case of online MDP problem, the hardness result is also applicable there.[1] The noisy parity problem is widely believed to be computationally intractable, and has been used to design cryptographic schemes.

**Organization**: In Section 3 we introduce an algorithm for MDP problems with adversarially chosen transition kernels and loss functions. Section 4 discusses how this algorithm can also be applied to the online episodic shortest path problem with adversarially varying graphs and loss functions and also considers the case of stochastic loss functions. Finally, in Section 4.2, we show the reduction from the adversarial online epsiodic shortest path problem to learning parity with noise.

## 2 Notations

Let $\mathcal{X} \subset \mathbb{R}^n$ be a state space and $\mathcal{A} \subset \mathbb{R}^d$ be an action space. Let $\Delta_S$ be the space of probability distributions over a set $S$. Define a policy $\pi$ as a mapping $\pi : \mathcal{X} \to \Delta_{\mathcal{A}}$. We use $\pi(a|x)$ to denote the probability of choosing an action $a$ in state $x$ under policy $\pi$. A random action under policy $\pi$ is denoted by $\pi(x)$. A transition probability kernel (or transition kernel) $m$ is a mapping $m : \mathcal{X} \times \mathcal{A} \to \Delta_{\mathcal{X}}$. For finite $\mathcal{X}$, let $P(\pi, m)$ be the transition probability matrix of policy $\pi$ under transition kernel $m$. A loss function is a bounded real-valued function over state and action spaces, $\ell : \mathcal{X} \times \mathcal{A} \to \mathbb{R}$. For a vector $v$, define $\|v\|_1 = \sum_i |v_i|$. For a real-valued function $f$ defined over $\mathcal{X} \times \mathcal{A}$, define $\|f\|_{\infty,1} = \max_{x \in \mathcal{X}} \sum_{a \in \mathcal{A}} |f(x,a)|$. The inner product between two vectors $v$ and $w$ is denoted by $\langle v, w \rangle$.

## 3 Online MDP Problems

In this section, we study a general MDP problem with large state and action spaces. The adversary can change the dynamics and the loss functions, but is restricted to choose dynamics that satisfy a mixing condition.

**Assumption A1 Uniform Mixing** There exists a constant $\tau > 0$ such that for all distributions $d$ and $d'$ over the state space, any deterministic policy $\pi$, and any transition kernel $m \in M$, $\|dP(\pi, m) - d'P(\pi, m)\|_1 \le e^{-1/\tau} \|d - d'\|_1$.

For all policies $\pi \in \Pi$, $w_{\pi,0} = 1$. $\eta = \min\{\sqrt{\log|\Pi|/T}, 1/2\}$.
Choose $\pi_1$ uniformly at random.
**for** $t := 1, 2, \ldots, T$ **do**
    Learner takes the action $a_t \sim \pi_t(.|x_t)$ and adversary chooses $m_t$ and $\ell_t$.
    Learner suffers loss $\ell_t(x_t, a_t)$ and observes $m_t$ and $\ell_t$. Update state: $x_{t+1} \sim m_t(.|x_t, a_t)$.
    For all policies $\pi$, $w_{\pi,t} = w_{\pi,t-1}(1-\eta)^{\mathbb{E}[\ell_t(x_t^\pi, \pi)]}$.
    $W_t = \sum_{\pi \in \Pi} w_{\pi,t}$. For any $\pi$, $p_{\pi,t+1} = w_{\pi,t}/W_t$.
    With probability $\beta_t = w_{\pi_t,t}/w_{\pi_t,t-1}$ choose the previous policy, $\pi_{t+1} = \pi_t$, while with
    probability $1 - \beta_t$, choose $\pi_{t+1}$ based on the distribution $p_{\pi,t+1}$.
**end for**

Figure 1: OMDP: The Online Algorithm for Markov Decision Processes

This assumption excludes deterministic MDPs that can be more difficult to deal with. As discussed by Neu et al. [14], if Assumption A1 holds for deterministic policies, then it holds for all policies.

We propose an exponentially-weighted average algorithm for this problem. The algorithm, called OMDP and shown in Figure 1, maintains a distribution over the policy class, but changes its policy with a small probability. The main results of this section are the following regret bounds for the OMDP algorithm. The proofs can be found in Appendix A.

**Theorem 1.** *Let the loss functions selected by the adversary be bounded in $[0, 1]$, and the transition kernels selected by the adversary satisfy Assumption A1. Then, for any policy $\pi \in \Pi$,*

$$\mathbb{E}\left[R_T(\text{OMDP}, \pi)\right] \leq (4 + 2\tau^2)\sqrt{T\log|\Pi|} + \log|\Pi| .$$

**Corollary 2.** *Let $\Pi$ be an arbitrary policy space, $\mathcal{N}(\epsilon)$ be the $\epsilon$-covering number of the metric space $(\Pi, \|.\|_{\infty,1})$, and $\mathcal{C}(\epsilon)$ be an $\epsilon$-cover. Assume that we run the OMDP algorithm on $\mathcal{C}(\epsilon)$. Then, under the same assumptions as in Theorem 1, for any policy $\pi \in \Pi$,*

$$\mathbb{E}\left[R_T(\text{OMDP}, \pi)\right] \leq (4 + 2\tau^2)\sqrt{T\log\mathcal{N}(\epsilon)} + \log\mathcal{N}(\epsilon) + \tau T\epsilon .$$

**Remark 3.** *If we choose $\Pi$ to be the space of deterministic policies and $\mathcal{X}$ and $\mathcal{A}$ are finite spaces, from Theorem 1 we obtain that $\mathbb{E}\left[R_T(\text{OMDP}, \pi)\right] \leq (4 + 2\tau^2)\sqrt{T|\mathcal{X}|\log|\mathcal{A}|} + |\mathcal{X}|\log|\mathcal{A}|$. This result, however, is not sufficient to show that the average regret with respect to the optimal stationary policy converges to zero. This is because, unlike in the standard MDP framework, the optimal stationary policy is not necessarily deterministic. Corollary 2 extends the result of Theorem 1 to continuous policy spaces.*

*In particular, if $\mathcal{X}$ and $\mathcal{A}$ are finite spaces and $\Pi$ is the space of all policies, $\mathcal{N}(\epsilon) \leq (|\mathcal{A}|/\epsilon)^{|\mathcal{A}||\mathcal{X}|}$, so the expected regret satisfies $\mathbb{E}\left[R_T(\text{OMDP}, \pi)\right] \leq (4+2\tau^2)\sqrt{T|\mathcal{A}||\mathcal{X}|\log\frac{|\mathcal{A}|}{\epsilon}} + |\mathcal{A}||\mathcal{X}|\log\frac{|\mathcal{A}|}{\epsilon} + \tau T\epsilon$. By the choice of $\epsilon = \frac{1}{T}$, we get that $\mathbb{E}\left[R_T(\text{OMDP}, \pi)\right] = O(\tau^2\sqrt{T|\mathcal{A}||\mathcal{X}|\log(|\mathcal{A}|T)})$.*

### 3.1 Proof Sketch

The main idea behind the design and the analysis of our algorithm is the following regret decomposition:

$$R_T(A, \pi) = \sum_{t=1}^{T} \ell_t(x_t^A, a_t) - \sum_{t=1}^{T} \ell_t(x_t^{\pi_t}, \pi_t) + \sum_{t=1}^{T} \ell_t(x_t^{\pi_t}, \pi_t) - \sum_{t=1}^{T} \ell_t(x_t^\pi, \pi) , \quad (1)$$

where $A$ is an online learning algorithm that generates a policy $\pi_t$ at round $t$, $x_t^A$ is the state at round $t$ if we have followed the policies generated by algorithm $A$, and $\ell(x, \pi) = \ell(x, \pi(x))$. Let

$$B_T(A) = \sum_{t=1}^{T} \ell_t(x_t^A, a_t) - \sum_{t=1}^{T} \ell_t(x_t^{\pi_t}, \pi_t) , \quad C_T(A, \pi) = \sum_{t=1}^{T} \ell_t(x_t^{\pi_t}, \pi_t) - \sum_{t=1}^{T} \ell_t(x_t^\pi, \pi) .$$

Note that the choice of policies has no influence over future losses in $C_T(A, \pi)$. Thus, $C_T(A, \pi)$ can be bounded by a reduction to full information online learning algorithms. Also, notice that the competitor policy $\pi$ does not appear in $B_T(A)$. In fact, $B_T(A)$ depends only on the algorithm $A$. We will show that if the class of transition kernels satisfies Assumption A1 and algorithm $A$ rarely changes its policies, then $B_T(A)$ can be bounded by a sublinear term. To be more precise, let $\alpha_t$ be the probability that algorithm $A$ changes its policy at round $t$. We will require that there exists a constant $D$ such that for any $1 \leq t \leq T$, any sequence of models $m_1, \ldots, m_t$ and loss functions $\ell_1, \ldots, \ell_t, \alpha_t \leq D/\sqrt{t}$.

We would like to have a full information online learning algorithm that rarely changes its policy. The first candidate that we consider is the well-known Exponentially Weighted Average (EWA) algorithm [15, 16]. In our MDP problem, the EWA algorithm chooses a policy $\pi \in \Pi$ according to distribution $q_t(\pi) \propto \exp\left(-\lambda \sum_{s=1}^{t-1} \mathbb{E}\left[\ell_s(x_s^\pi, \pi)\right]\right)$ for some $\lambda > 0$. The policies that this EWA algorithm generates most likely are different in consecutive rounds and thus, the EWA algorithm might change its policy frequently. However, a variant of EWA, called Shrinking Dartboard (SD) (Guelen et al. [17]), rarely changes its policy (see Lemma 8 in Appendix A). The OMDP algorithm is based on the SD algorithm. Note that the algorithm needs to know the number of rounds, $T$, in advance. Also note that we could use any rarely switching algorithm such as Follow the Lazy Leader of Kalai and Vempala [18] as the subroutine.

## 4 Adversarial Online Shortest Path Problem

We consider the following adversarial online shortest path problem with changing graphs. The problem is a repeated game played between a decision-maker and an (oblivious) adversary over $T$ rounds. At each round $t$ the adversary presents a directed acyclic graph $g_t$ on $n$ nodes to the decision maker, with $L$ layers indexed by $\{1, \ldots, L\}$ and a special *start* and *finish* node. Each layer contains a fixed set of nodes and has connections only with the next layer. [2] The decision-maker must choose a path $p_t$ from the start to the finish node. Then, the adversary reveals weights across all the edges of the graph. The *loss* $\ell_t(g_t, p_t)$ of the decision-maker is the weight along the path that the decision-maker took on that round.

Denote by $[k]$ the set $\{1, 2, \ldots, k\}$. A *policy* is a mapping $\pi : [n] \to [n]$. Each policy may be interpreted as giving a start to finish path. Suppose that the start node is $s \in [n]$, then $\pi(i)$ gives the subsequent node. The path is interpreted as follows : if at a node $v$, the edge $(v, \pi(v))$ exists then the next node is $\pi(v)$. Otherwise, the next node is an arbitrary (pre-determined) choice that is adjacent to $v$. We compete against the class of such policies for choosing the shortest path. Denote the class of such policies by $\Pi$. The regret of a decision-maker $A$ with respect to a policy $\pi \in \Pi$ is defined as: $R_T(A, \pi) = \sum_{t=1}^{T} \ell_t(g_t, p_t) - \sum_{t=1}^{T} \ell_t(g_t, \pi(g_t))$, where $\pi(g_t)$ is the path obtained by following the policy $\pi$ starting at the source node. Note that it is possible that there exists no policy that would result in an actual path that leads to the sink for some graph. In this case we say that the loss of the policy is infinite. Thus, there may be adversarially chosen sequences of graphs for which the regret of a decision-maker is not well-defined. This can be easily corrected by the adversary ensuring that the graph always has some *fixed* set of edges which result in a (possibly high loss) $s \to f$ path. In fact, we show that the adversary can choose a sequence of graphs and loss functions that make this problem at least as hard as learning noisy parities. Learning noisy parities is a notoriously hard problem in computational learning theory. The best known algorithm runs in time $2^{O(n/\log(n))}$ [20] and the presumed hardness of this and related problems has been used for designing cryptographic protocols [21].

Interestingly, for the hardness result to hold, it is essential that the adversary have the ability to control both the sequence of graphs and losses. The problem is well-understood when the graphs are generated randomly and the losses are adversarial. Jaksch et al. [3] and Bartlett and Tewari [4] propose efficient algorithms for problems with stochastic losses.[3] Neu et al. [22] extend these results to problems with adversarial loss functions.

One can also ask what happens in the case when the graphs are chosen by the adversary, but the weight of each edge is drawn at random according to a fixed stationary distribution. In this setting, we show a reduction to bandit linear optimization. Thus, in fact, that algorithm does not need to see the weights of all edges at the end of the round, but only needs to know the loss it suffered.

Finally, we consider the case when both graphs and losses are chosen adversarially. Although the general problem is at least as hard as learning noisy parities, we give an efficient algorithm whose regret scales linearly with the number of different graphs. Thus, if the adversary is forced to choose graphs from some small set $\mathcal{G}$, then we have an efficient algorithm for solving the problem. We note that in fact, our algorithm does not need to see the graph $g_t$ at the beginning of the round, in which case an algorithm achieving $O(|\mathcal{G}|\sqrt{T})$ may be trivially obtained.

## 4.1 Stochastic Loss Functions and Adversarial Graphs

Consider the case when the weight of each edge is chosen from a fixed distribution. Then it is easy to see that the expected loss of any path is a fixed linear function of the expected weights vector. The set of available paths depends on the graph and it may change from time to time. This is an instance of stochastic linear bandit problem, for which efficient algorithms exist [23, 24, 25].

**Theorem 4.** *Let us represent each path by a binary vector of length $n(n-1)/2$, such that the ith element is 1 only if the corresponding edge is present in the path. Assume that the learner suffers the loss of $c(p)$ for choosing path $p$, where $\mathbb{E}\left[c(p)\right] = \langle \ell, p \rangle$ and the loss vector $\ell \in \mathbb{R}^{n(n-1)/2}$ is fixed. Let $P_t$ be the set of paths in a graph $g_t$. Consider the CONFIDENCEBALL$_1$ algorithm of Dani et al. [24] applied to the shortest path problem with a changing action set $P_t$ and the loss function $\ell$. Then the regret with respect to the best path in each round is $Cn^3\sqrt{T}$ for a problem-independent constant $C$.*

Let $\widehat{\ell}_t$ be the least squares estimate of $\ell$ at round $t$, $V_t = \sum_{s=1}^{t-1} p_s p_s^\top$ be the covariance matrix, and $P_t$ be the decision set at round $t$. The CONFIDENCEBALL$_1$ algorithm constructs a high probability norm-1 ball confidence set, $C_t = \left\{ \ell \ : \ \left\| V_t^{1/2}(\ell - \widehat{\ell}_t) \right\|_1 \leq \beta_t \right\}$ for an appropriate $\beta_t$, and chooses an action $p_t$ according to $p_t = \operatorname{argmin}_{\ell \in C_t, p \in P_t} \langle \ell, p \rangle$. Dani et al. [24] prove that the regret of the CONFIDENCEBALL$_1$ algorithm is bounded by $O(m^{3/2}\sqrt{T})$, where $m$ is the dimensionality of the action set (in our case $m = n(n-1)/2$). The above optimization can be solved efficiently, because only $2n$ corners of $C_t$ need to be evaluated.

Note that the regret in Theorem 4 is with respect to the best path in each round, which is a stronger result than competing with a fixed policy.

## 4.2 Hardness Result

In this section, we show that the setting when both the graphs and the losses are chosen by an adversary, the problem is at least as hard as the noisy parity problem. We consider the online agnostic parity learning problem. Recall that the class of parity function over $\{0,1\}^n$ is the following: For $S \subseteq [n]$, $\mathsf{PAR}_S(x) = \oplus_{i \in S} x_i$, where $\oplus$ denotes modulo 2 addition. The class is $\mathsf{PARITIES} = \{\mathsf{PAR}_S \mid S \subseteq [n]\}$. In the online setting, the learning algorithm is given $x^t \in \{0,1\}^n$, the learning algorithm then picks $\hat{y}^t \in \{0,1\}$, and then the true label $y^t$ is revealed. The learning algorithm suffers loss $\mathbb{I}(\hat{y}^t \neq y^t)$. The regret of the learning algorithm with respect to $\mathsf{PARITIES}$ is defined as: $\mathrm{Regret} = \sum_{t=1}^T \mathbb{I}(\hat{y}^t \neq y^t) - \min_{\mathsf{PAR}_S \in \mathsf{PARITIES}} \sum_{t=1}^T \mathbb{I}(\mathsf{PAR}_S(x^t) \neq y^t)$. The goal is to design a learning algorithm that runs in time polynomial in $n, T$ and suffers regret $O(\operatorname{poly}(n)T^{1-\delta})$ for some constant $\delta > 0$. It follows from prior work that online agnostic learning of parities is at least as hard as the offline version (see Littlestone [26], Kanade and Steinke [27]). As mentioned previously, the agnostic parity learning problem is notoriously difficult. Thus, it seems unlikely that a *computationally efficient* no-regret algorithm for this problem exists.

**Theorem 5.** *Suppose there is a no-regret algorithm for the online adversarial shortest path problem that runs in time $\operatorname{poly}(n,T)$ and achieves regret $O(\operatorname{poly}(n)T^{1-\delta})$ for any constant $\delta > 0$. Then there is a polynomial-time algorithm for online agnostic parity learning that achieves regret $O(\operatorname{poly}(n)T^{1-\delta})$. By the online to batch reduction, this would imply a polynomial time algorithm for agnostically learning parities.*

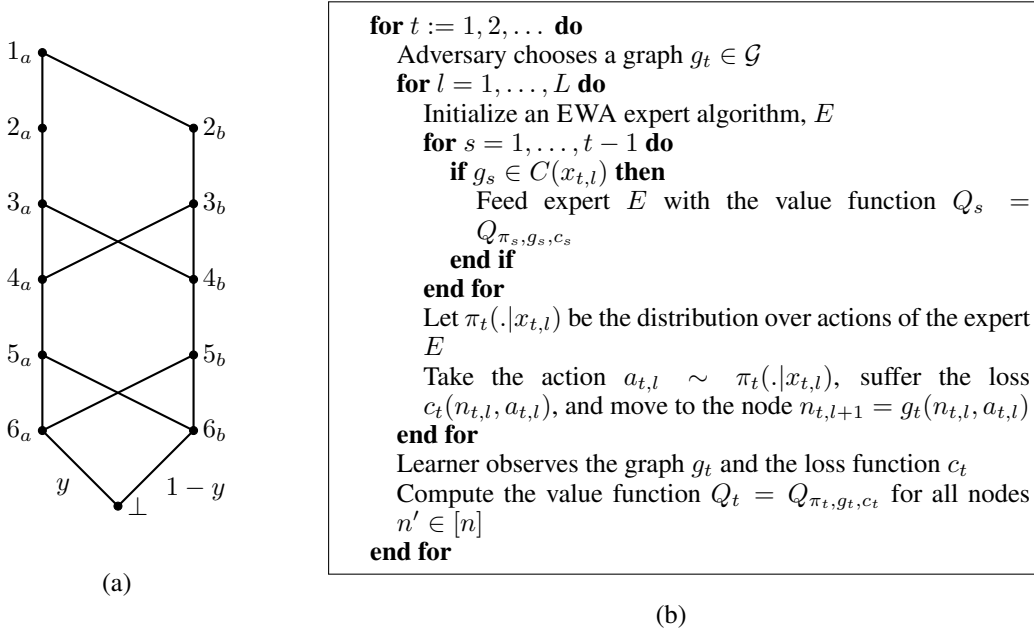

(a)

```
for t := 1, 2, ... do
    Adversary chooses a graph g_t ∈ G
    for l = 1, ..., L do
        Initialize an EWA expert algorithm, E
        for s = 1, ..., t − 1 do
            if g_s ∈ C(x_{t,l}) then
                Feed expert E with the value function Q_s = Q_{π_s, g_s, c_s}
            end if
        end for
        Let π_t(.|x_{t,l}) be the distribution over actions of the expert E
        Take the action a_{t,l} ∼ π_t(.|x_{t,l}), suffer the loss c_t(n_{t,l}, a_{t,l}), and move to the node n_{t,l+1} = g_t(n_{t,l}, a_{t,l})
    end for
    Learner observes the graph g_t and the loss function c_t
    Compute the value function Q_t = Q_{π_t, g_t, c_t} for all nodes n' ∈ [n]
end for
```

(b)

Figure 2: (a) Encoding the example $(1, 0, 1, 0, 1) \in \{0, 1\}^5$ as a graph. (b) Improved Algorithm for the Online Shortest Path Problem.

*Proof.* We first show how to map a point $(x, y)$ to a graph and a loss function. Let $(x, y) \in \{0, 1\}^n \times \{0, 1\}$. We define a graph, $g(x)$ and a loss function $\ell_{x,y}$ associated with $(x, y)$. Define a graph on $2n + 2$ nodes – named $1_a, 2_a, 2_b, 3_a, 3_b, \ldots, n_a, n_b, (n + 1)_a, (n + 1)_b, \bot$ in that order. Let $E(x)$ denote the set of edges of $g(x)$. The set $E(x)$ contains the following edges:

(i) If $x_1 = 1$, both $(1_a, 2_a)$ and $(1_a, 2_b)$ are in $E(x)$, else if $x_1 = 0$, only $(1_a, 2_a)$ is present.

(ii) For $1 < i \leq n$, if $x_i = 1$, the edges $(i_a, (i + 1)_a), (i_a, (i + 1)_b), (i_b, (i + 1)_a), (i_b, (i + 1)_b)$ are all present; if $x_i = 0$ only the two edges $(i_a, (i + 1)_a)$ and $(i_b, (i + 1)_b)$ are present.

(iii) The two edges $((n + 1)_a, \bot)$ and $((n + 1)_b, \bot)$ are always present.

For the loss function, define the weights as follows. The weight of the edge $((n + 1)_a, \bot)$ is $y$; the weight of the edge $((n + 1)_b, \bot)$ is $1 - y$. The weights of all the remaining edges are set to 0. Figure 2(a) shows the encoding of the example $(1, 0, 1, 0, 1) \in \{0, 1\}^5$.

Suppose an algorithm with the stated regret bound for the online shortest path problem exists, call it $\mathcal{U}$. We will use this algorithm to solve the online parity learning problem. Let $x^t$ be an example received; then pass the graph $g(x^t)$ to the algorithm $\mathcal{U}$. The start vertex is $1_a$ and the finish vertex is $\bot$. Suppose the path $p^t$ chosen by $\mathcal{U}$ reaches $\bot$ using the edge $((n + 1)_a, \bot)$ then set $\hat{y}^t$ to be 0. Otherwise, choose $\hat{y}^t = 1$.

Thus, in effect we are using algorithm $\mathcal{U}$ as a meta-algorithm for the online agnostic parity learning problem. First, it is easy to check that the loss suffered by the *meta-algorithm* on the parity problem is exactly the same as the loss of $\mathcal{U}$ on the online shortest path problem. This follows directly from the definition of the losses on the edges.

Next, we claim that for any $S \subseteq [n]$, there is a policy $\pi_S$ that achieves the same loss (on the online shortest path problem) as the parity $\mathsf{PAR}_S$ does (on the parity learning problem). The policy is as follows:

(i) From node $i_a$, if $i \in S$ and $(i_a, (i + 1)_b) \in E(g^t)$, go to $(i + 1)_b$, otherwise go to $(i + 1)_a$.

(ii) From node $i_b$, if $i \in S$ and $(i_b, (i + 1)_a) \in E(g^t)$, go to $(i + 1)_a$, otherwise go to $(i + 1)_b$.

(iii) Finally, from either $(n + 1)_a$ or $(n + 1)_b$, just move to $\bot$.

We can think of the path $p^t$ as being in type $a$ nodes or type $b$ nodes. For each $i \in S$, such that $x_i^t = 1$, the path $p^t$ switches types. Thus, if $\mathsf{PAR}_S(x^t) = 1$, $p^t$ reaches $\bot$ via the edge $((n + 1)_b, \bot)$ and if $\mathsf{PAR}_S(x^t) = 0$, $p^t$ reaches $\bot$ via the edge $((n + 1)_a, \bot)$. Recall that the loss function is

defined as follows: weight of the edge $((n+1)_a, \perp)$ is $y^t$, weight of the edge $((n+1)_b, \perp)$ is $1 - y^t$; other edges have loss 0. Thus, the loss suffered by the policy $\pi_S$ is 1 if $\mathsf{PAR}_S(x^t) \neq y^t$ and 0 otherwise. This is exactly the loss of the parity function $\mathsf{PAR}_S$ on the agnostic parity learning problem. Thus, if the algorithm $\mathcal{U}$ has regret $O(\text{poly}(n), T^{1-\delta})$, then the meta-algorithm for the online agnostic parity learning problem also has regret $O(\text{poly}(n), T^{1-\delta})$. $\qquad\square$

**Remark 6.** *We observe that the online shortest path problem is a special case of online MDP learning. Thus, the above reduction also shows that, short of a major breakthrough, it is unlikely that there exists a computationally efficient algorithm for the fully adversarial online MDP problem.*

### 4.3 Small Number of Graphs

In this section, we design an efficient algorithm and prove a $O(|\mathcal{G}| \sqrt{T})$ regret bound, where $\mathcal{G}$ is the set of graphs played by the adversary up to round $T$. The computational complexity of the algorithm is $O(L^2 t)$ at round $t$. The algorithm does not need to know the set $\mathcal{G}$ or $|\mathcal{G}|$. This regret bound holds even if the graphs are revealed at the end of the rounds. Notice that if the graphs are shown at the beginning of the rounds, obtaining regret bounds that scale like $O(|\mathcal{G}| \sqrt{T})$ is trivial; the learner only needs to run $|\mathcal{G}|$ copies of the MDP-E algorithm of Even-Dar et al. [12], one for each graph.

Let $n_{t,l}^\pi$ denote the node at layer $l$ of round $t$ if we run policy $\pi$. Let $c_t(n', a)$ be the loss incurred for taking action $a$ in node $n'$ at round $t$.[4] We construct a new graph, called $G$, as follows: graph $G$ also has a layered structure with the same number of layers, $L$. At each layer, we have a number of *states* that represent all possible observations that we might have upon arriving at that layer. Thus, a state at layer $l$ has the form of $x = (s, a_0, n_1, a_1, \ldots, n_{l-1}, a_{l-1}, n_l)$, where $n_i$ belongs to layer $i$ and $a_i \in \mathcal{A}$.

Let $\mathcal{X}$ be the set of states in $G$ and $\mathcal{X}_l$ be the set of states in layer $l$ of $G$. For $(x, a) \in \mathcal{X} \times \mathcal{A}$, let $c(x, a) = c(n(x), a)$, where $n(x)$ is the last node observed in state $x$. Let $g(n', a)$ be the next node under graph $g$ if we take action $a$ in node $n'$. Let $g(x, a) = g(n(x), a)$. Let $c(x, \pi) = \sum_a \pi(a|x) c(x, a)$. For a graph $g$ and a loss function $\ell$, define the value functions by

$$\forall n' \in [n], \quad Q_{\pi,g,c}(n', \pi') = \mathbb{E}_{a \sim \pi'(n')} \left[ c(n', a) + Q_{\pi,g,c}(g(n', a), \pi) \right],$$

$$\forall x, \text{s.t. } g \in C(x), \quad Q_{\pi,g,c}(x, \pi') = Q_{\pi,g,c}(n(x), \pi'),$$

with $Q_{\pi,g,c}(f, a) = 0$ for any $\pi, g, c, a$ where $f$ is the finish node. Let $Q_t = Q_{\pi_t, g_t, c_t}$ denote the value function associated with policy $\pi_t$ at time $t$. For $x = (s, a_0, n_1, a_1, \ldots, n_{l-1}, a_{l-1}, n_l)$, define $C(x) = \{g \in \mathcal{G} : n_1 = g(s, a_0), \ldots, n_l = g(n_{l-1}, a_{l-1})\}$, the set of graphs that are consistent with the state $x$.

We can use the MDP-E algorithm to generate policies. The algorithm, however, is computationally expensive as it updates a large set of experts at each round. Notice that the number of states at stage $l$, $|\mathcal{X}_l|$, can be exponential in the number of graphs. We show a modification of the MDP-E algorithm that would generate the same sequence of policies, with the advantage that the new algorithm is computationally efficient. The algorithm is shown in Figure 2(b). As the generated policies are always the same, the regret bound in the next theorem, that is proven for the MDP-E algorithm, also applies to the new algorithm. The proof can be found in Appendix B.

**Theorem 7.** *For any policy* $\pi$, $\mathbb{E}\left[R_T(\text{MDP-E}, \pi)\right] \leq 2L\sqrt{8T \log(2T)} + L \min\{|\mathcal{G}|, \max_l |\mathcal{X}_l|\} \sqrt{T \log \frac{|\mathcal{A}|}{2}} + 2L.$

The theorem gives a sublinear regret as long as $|\mathcal{G}| = o(\sqrt{T})$. On the other hand, the hardness result in Theorem 5 applies when $|\mathcal{G}| = \Theta(T)$. Characterizing regret vs. computational complexity tradeoffs when $|\mathcal{G}|$ is in between remains for future work.

## Footnotes

[1]There was an error in the proof of a claimed hardness result for the online adversarial MDP problem [8]; this claim has since been retracted [12, 13].

[2]As noted by Neu et al. [19], any directed acyclic graph can be transformed into a graph that satisfies our assumptions.

[3]These algorithms are originally proposed for continuing problems, but we can use them in shortest path problems with small modifications.

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
