[Supplementary Material]

# A  Proofs of Section 3

Consider a basic full information problem with $N$ experts. Let $R_T(\text{SD}, i)$ be the regret of the SD algorithm with respect to expert $i$ up to time $T$. We have the following results for the SD algorithm.

**Lemma 8.** *For any expert $i \in \{1, \ldots, N\}$, $R_T(\text{SD}, i) \leq 4\sqrt{T \log N} + \log N$, and also for any $1 \leq t \leq T$, $\mathbb{P}\left(\text{Switch at time } t\right) \leq \sqrt{\frac{\log N}{T}}$.*

*Proof.* The proof of the regret bound can be found in [17, Theorem 3]. The proof of the bound on the probability of switch is similar to the proof of Lemma 2 in [17] and is as follows: As shown in [17, Lemma 2], the probability of switch at time $t$ is $\alpha_t = (W_{t-1} - W_t)/W_{t-1}$. Thus, $W_t = (1 - \alpha_t)W_{t-1}$. Because the loss function is bounded in $[0, 1]$, we have that

$$W_t = \sum_{i=1}^{N} w_{i,t} = \sum_{i=1}^{N} w_{i,t-1}(1 - \eta)^{c_t(i)} \geq \sum_{i=1}^{N} w_{i,t-1}(1 - \eta) = (1 - \eta)W_{t-1} \ .$$

Thus, $1 - \alpha_t \geq 1 - \eta$, and thus, $\alpha_t \leq \eta \leq \sqrt{(\log N)/T}$.  $\square$

## A.1  Proof of Theorem 1

In the rest of this section, we write $A$ to denote the OMDP algorithm. For the proof we use the regret decomposition (1):
$$R_T(A, \pi) = B_T(A) + C_T(A, \pi) \ .$$

**Lemma 9.** *For any policy $\pi \in \Pi$,*

$$\mathbb{E}\left[C_T(A, \pi)\right] = \mathbb{E}\left[\sum_{t=1}^{T} \ell_t(x_t^{\pi_t}, \pi_t) - \sum_{t=1}^{T} \ell_t(x_t^{\pi}, \pi)\right] \leq 4\sqrt{T \log |\Pi|} + \log |\Pi| \ .$$

*Proof.* Consider the following imaginary game between a learner and an adversary: we have a set of experts (policies) $\Pi = \{\pi^1, \ldots, \pi^{|\Pi|}\}$. At round $t$, the adversary chooses a loss vector $c_t \in [0, 1]^{\Pi}$, whose $i$th element determines the loss of expert $\pi^i$ at this round. The learner chooses a distribution over experts $q_t$ (defined by the SD algorithm), from which it draws an expert $\pi_t$. Next, the learner observes the loss function $c_t$. From the regret bound for the SD algorithm (Lemma 8), it is guaranteed that for any expert $\pi$,

$$\sum_{t=1}^{T} \langle c_t, q_t \rangle - \sum_{t=1}^{T} c_t(\pi) \leq 4\sqrt{T \log |\Pi|} + \log |\Pi| \ .$$

Next, we determine how the adversary chooses the loss vector. At time $t$, the adversary chooses a loss function $\ell_t$ and sets $c_t(\pi^i) = \mathbb{E}\left[\ell_t(x_t^{\pi^i}, \pi^i)\right]$. Noting that $\langle c_t, q_t \rangle = \mathbb{E}\left[\ell_t(x_t^{\pi_t}, \pi_t)\right]$ and $c_t(\pi) = \mathbb{E}\left[\ell_t(x_t^{\pi}, \pi)\right]$ finishes the proof.  $\square$

**Lemma 10.** *We have that*

$$\mathbb{E}\left[B_T(A)\right] = \mathbb{E}\left[\sum_{t=1}^{T} \ell_t(x_t^A, a_t) - \sum_{t=1}^{T} \ell_t(x_t^{\pi_t}, \pi_t)\right] \leq 2\tau^2 \sqrt{\log |\Pi| T} \ .$$

First, we state the following two lemmas.

**Lemma 11** (Lemma 5.1 of Even-Dar et al. [12])**.** *For any state distribution $d$, any transition kernel $m$, and any policies $\pi$ and $\pi'$,*

$$\|dP(\pi, m) - dP(\pi', m)\|_1 \leq \|\pi - \pi'\|_{\infty,1} \ .$$

Note that matrix $P(\pi, m)$ was defined for finite state spaces, but with appropriate modifications the same argument works for continuous state spaces as well.

**Lemma 12.** *Let $\alpha_t$ be the probability of a policy switch at time $t$. Then, $\alpha_t \leq \sqrt{\log |\Pi|/T}$.*

*Proof.* Proof is identical to the proof of Lemma 8. □

*Proof of Lemma 10.* Let $\mathcal{F}_t = \sigma(\pi_1, \ldots, \pi_t)$. Notice that the choice of policies are independent of the state variables. We can write

$$
\begin{aligned}
\mathbb{E}\left[B_T(A)\right] &= \mathbb{E}\left[\sum_{t=1}^{T} \ell_t(x_t^A, a_t) - \sum_{t=1}^{T} \ell_t(x_t^{\pi_t}, \pi_t)\right] \\
&= \mathbb{E}\left[\sum_{t=1}^{T} \sum_{x \in \mathcal{X}} \left(\mathbb{I}_{\{x_t^A=x\}} - \mathbb{I}_{\{x_t^{\pi_t}=x\}}\right) \ell_t(x, \pi_t(x))\right] \\
&= \mathbb{E}\left[\sum_{t=1}^{T} \sum_{x \in \mathcal{X}} \mathbb{E}\left[\left(\mathbb{I}_{\{x_t^A=x\}} - \mathbb{I}_{\{x_t^{\pi_t}=x\}}\right) \ell_t(x, \pi_t(x)) \,\Big|\, \mathcal{F}_T\right]\right] \\
&= \mathbb{E}\left[\sum_{t=1}^{T} \sum_{x \in \mathcal{X}} \ell_t(x, \pi_t(x)) \mathbb{E}\left[\left(\mathbb{I}_{\{x_t^A=x\}} - \mathbb{I}_{\{x_t^{\pi_t}=x\}}\right) \,\Big|\, \mathcal{F}_T\right]\right] \\
&\leq \mathbb{E}\left[\sum_{t=1}^{T} \|\ell_t\|_\infty \left\|\mathbb{E}\left[\left(\mathbb{I}_{\{x_t^A=x\}} - \mathbb{I}_{\{x_t^{\pi_t}=x\}}\right) \,\Big|\, \mathcal{F}_T\right]\right\|_1\right] \\
&= \mathbb{E}\left[\sum_{t=1}^{T} \|\ell_t\|_\infty \|u_t - v_{t,t}\|_1\right] \\
&\leq \mathbb{E}\left[\sum_{t=1}^{T} \|u_t - v_{t,t}\|_1\right],
\end{aligned}
\tag{2}
$$

where $u_s = \mathbb{E}\left[\mathbb{I}_{\{x_s^A=x\}} \big| \mathcal{F}_T\right]$ is the distribution of $x_s^A$ for $s \leq t$ and $v_{s,t} = \mathbb{E}\left[\mathbb{I}_{\{x_s^{\pi_t}=x\}} \big| \mathcal{F}_T\right]$ is the distribution of $x_s^{\pi_t}$ for $s \leq t$.[5] Let $E_t$ be the event of a policy switch at time $t$. From inequality

$$
\|\pi_{t-k} - \pi_t\|_{\infty,1} \leq \|\pi_{t-k} - \pi_{t-k+1}\|_{\infty,1} + \cdots + \|\pi_{t-1} - \pi_t\|_{\infty,1} \leq 2 \sum_{s=t-k+1}^{t} \mathbb{I}_{\{E_s\}},
$$

and Lemma 12, we get that

$$
\mathbb{E}\left[\|\pi_{t-k} - \pi_t\|_{\infty,1}\right] \leq 2\sqrt{\frac{\log |\Pi|}{T}} k .
\tag{3}
$$

Let $P_t^\pi = P(\pi, m_t)$. We have that

$$
\begin{aligned}
\mathbb{E}\left[\|u_t - v_{t,t}\|_1\right] &= \mathbb{E}\left[\left\|u_{t-1}P_{t-1}^{\pi_{t-1}} - v_{t-1,t}P_{t-1}^{\pi_t}\right\|_1\right] \\
&= \mathbb{E}\left[\left\|u_{t-1}P_{t-1}^{\pi_{t-1}} - u_{t-1}P_{t-1}^{\pi_t} + u_{t-1}P_{t-1}^{\pi_t} - v_{t-1,t}P_{t-1}^{\pi_t}\right\|_1\right] \\
&\leq \mathbb{E}\left[\left\|u_{t-1}P_{t-1}^{\pi_{t-1}} - u_{t-1}P_{t-1}^{\pi_t}\right\|_1 + \left\|u_{t-1}P_{t-1}^{\pi_t} - v_{t-1,t}P_{t-1}^{\pi_t}\right\|_1\right] \\
&\leq \mathbb{E}\left[\|\pi_{t-1} - \pi_t\|_{\infty,1} + e^{-1/\tau}\|u_{t-1} - v_{t-1,t}\|_1\right] \\
&\leq \mathbb{E}\Big[\|\pi_{t-1} - \pi_t\|_{\infty,1} + e^{-1/\tau}(\left\|u_{t-2}P_{t-2}^{\pi_{t-2}} - u_{t-2}P_{t-2}^{\pi_t}\right\|_1 \\
&\qquad\qquad + \left\|u_{t-2}P_{t-2}^{\pi_t} - v_{t-2,t}P_{t-2}^{\pi_t}\right\|_1)\Big] \\
&\leq \mathbb{E}\left[\|\pi_{t-1} - \pi_t\|_{\infty,1} + e^{-1/\tau}\|\pi_{t-2} - \pi_t\|_{\infty,1} + e^{-2/\tau}\|u_{t-2} - v_{t-2,t}\|_1\right] \\
&\leq \ldots \\
&\leq \sum_{k=0}^{t} e^{-k/\tau}\mathbb{E}\left[\|\pi_{t-k} - \pi_t\|_{\infty,1}\right] + e^{-t/\tau}\|u_0 - v_{0,t}\|_1 \\
&\leq \sum_{k=0}^{t} 2e^{-k/\tau}\sqrt{\frac{\log|\Pi|}{T}}k + 0 \qquad \text{By (3)} \\
&\leq 2\sqrt{\frac{\log|\Pi|}{T}}\tau^2 , \tag{4}
\end{aligned}
$$

where we have used the fact that $\|u_0 - v_{0,t}\|_1 = 0$, because the initial distributions are identical. By (4) and (2), we get that

$$
\mathbb{E}\left[B_T(A)\right] \leq 2\tau^2 \sum_{t=1}^{T} \sqrt{\frac{\log|\Pi|}{T}} = 2\tau^2\sqrt{\log|\Pi|T} .
$$

$\qquad\square$

What makes the analysis possible is the fact that all policies mix no matter what transition kernel is played by the adversary.

*Proof of Theorem 1.* The result is obvious by Lemmas 9 and 10. $\qquad\square$

## A.2 Proof of Corollary 2

*Proof of Corollary 2.* Let $L_T(\pi) = \mathbb{E}\left[\sum_{t=1}^{T}\ell_t(x_t^\pi, \pi)\right]$ be the value of policy $\pi$. Let $u_{\pi,t}(x) = \mathbb{P}(x_t^\pi = x)$. First, we prove that the value function is Lipschitz with Lipschitz constant $\tau T$. The argument is similar to the argument in the proof of Lemma 10. For any $\pi_1$ and $\pi_2$,

$$
\begin{aligned}
|L_T(\pi_1) - L_T(\pi_2)| &= \left|\mathbb{E}\left[\sum_{t=1}^{T}\ell_t(x_t^{\pi_1}, \pi_1) - \sum_{t=1}^{T}\ell_t(x_t^{\pi_2}, \pi_2)\right]\right| \leq 2\left|\sum_{t=1}^{T}\|u_{\pi_1,t} - u_{\pi_2,t}\|_1\|\ell_t\|_\infty\right| \\
&\leq 2\left|\sum_{t=1}^{T}\|u_{\pi_1,t} - u_{\pi_2,t}\|_1\right| .
\end{aligned}
$$

With an argument similar to the one in the proof of Lemma 10, we can show that $\|u_{\pi_1,t} - u_{\pi_2,t}\|_1 \leq \tau\|\pi_1 - \pi_2\|_{\infty,1}$. Thus, $|L_T(\pi_1) - L_T(\pi_2)| \leq \tau T\|\pi_1 - \pi_2\|_{\infty,1}$. Given this and the fact that for any policy $\pi \in \Pi$, there is a policy $\pi' \in \mathcal{C}(\epsilon)$ such that $\|\pi - \pi'\|_{\infty,1} \leq \epsilon$, we get that

$$
\mathbb{E}\left[R_T(\text{OMDP}, \pi)\right] \leq (4 + 2\tau^2)\sqrt{T\log\mathcal{N}(\epsilon)} + \log\mathcal{N}(\epsilon) + \tau T\epsilon .
$$

$\qquad\square$

# B   Proof of Theorem 7

Let $x_{t,l}$ and $a_{t,l}$ denote the state and the action at step $l$ of episode $t$. Let $x_{t,l}^{\pi}$ denote the state at stage $l$ of round $t$ if we run policy $\pi$. As $c(x,a) = c(n(x),a)$, we can write that

$$R_T(\pi) = \sum_{t=1}^{T}\sum_{l=1}^{L} c_t(x_{t,l}^{\pi_t}, \pi_t) - \sum_{t=1}^{T}\sum_{l=1}^{L} c_t(x_{t,l}^{\pi}, \pi) \; .$$

We have the following regret decomposition:

$$R_T(\pi) = B_T + C_T \; ,$$

where

$$B_T = \sum_{t=1}^{T}\sum_{l=1}^{L}\left( c_t(x_{t,l}^{\pi_t}, \pi_t) - Q_t(s, \pi_t)/L \right) \; , \quad C_T = \sum_{t=1}^{T}\sum_{l=1}^{L}\left( Q_t(s, \pi_t)/L - c_t(x_{t,l}^{\pi}, \pi) \right) \; .$$

We bound these terms in the following sections.

## B.1   Bounding $\mathbb{E}\left[C_T\right]$

First, we prove the following lemma.

**Lemma 13.** *Let $\pi$ be a policy. Let $x = (s, a_0, n_1, a_1, \dots, n_{l-1}, a_{l-1}, n_l)$. We have that*

$$\sum_{x \in \mathcal{X}_l} \pi(a_0|s)\dots\pi(a_{l-1}|n_{l-1}) \le |\mathcal{G}| \; .$$

*Proof.* For any graph $g$,

$$\sum_{x:g\in C(x)} \pi(a_0|s)\dots\pi(a_{l-1}|n_{l-1}) = 1 \; .$$

We get the result by summing over the graphs.   $\square$

**Lemma 14.** *We have that*

$$\mathbb{E}\left[C_T\right] \le L\,|\mathcal{G}|\sqrt{T\log\frac{|\mathcal{A}|}{2}} + L\sqrt{8T\log(2T)} + L \; .$$

*Proof.* For any step $l$ during an episode $t$, we have that $Q_t(x_{t,l}^{\pi}, \pi) = c_t(x_{t,l}^{\pi}, \pi) + \mathbb{E}\left[Q_t(x_{t,l+1}^{\pi}, \pi_t) \,|\, x_{t,l}^{\pi}\right]$. Thus,

$$Q_t(x_{t,l}^{\pi}, \pi) = c_t(x_{t,l}^{\pi}, \pi) + \mathbb{E}\left[Q_t(x_{t,l+1}^{\pi}, \pi_t)\,|\,x_{t,l}^{\pi}\right] - \mathbb{E}\left[Q_t(x_{t,l}^{\pi}, \pi_t)\,|\,x_{t,l-1}^{\pi}\right]$$
$$+ \mathbb{E}\left[Q_t(x_{t,l}^{\pi}, \pi_t)\,|\,x_{t,l-1}^{\pi}\right] - Q_t(x_{t,l}^{\pi}, \pi_t) + Q_t(x_{t,l}^{\pi}, \pi_t) \; .$$

For episode $t$,

$$\sum_{l=1}^{L}\left( Q_t(x_{t,l}^{\pi}, \pi) - Q_t(x_{t,l}^{\pi}, \pi_t) \right) = \sum_{l=1}^{L} c_t(x_{t,l}^{\pi}, \pi)$$

$$+ \sum_{l=1}^{L}\left( \mathbb{E}\left[Q_t(x_{t,l+1}^{\pi}, \pi_t)\,|\,x_{t,l}^{\pi}\right] - \mathbb{E}\left[Q_t(x_{t,l}^{\pi}, \pi_t)\,|\,x_{t,l-1}^{\pi}\right] \right)$$

$$+ \sum_{l=1}^{L}\left( \mathbb{E}\left[Q_t(x_{t,l}^{\pi}, \pi_t)\,|\,x_{t,l-1}^{\pi}\right] - Q_t(x_{t,l}^{\pi}, \pi_t) \right)$$

$$= -Q_t(s, \pi_t) + \sum_{l=1}^{L} c_t(x_{t,l}^{\pi}, \pi)$$

$$+ \sum_{l=1}^{L}\left( \mathbb{E}\left[Q_t(x_{t,l}^{\pi}, \pi_t)\,|\,x_{t,l-1}^{\pi}\right] - Q_t(x_{t,l}^{\pi}, \pi_t) \right) \; .$$

Thus,

$$\sum_{t=1}^{T}\sum_{l=1}^{L}\left(Q_t(s,\pi_t)/L - c_t(x_{t,l}^\pi,\pi)\right) = \sum_{t=1}^{T}\sum_{l=1}^{L}\left(Q_t(x_{t,l}^\pi,\pi_t) - Q_t(x_{t,l}^\pi,\pi)\right)$$

$$+ \sum_{l=1}^{L}\left(\mathbb{E}\left[Q_t(x_{t,l}^\pi,\pi_t)\,|\,x_{t,l-1}^\pi\right] - Q_t(x_{t,l}^\pi,\pi_t)\right) \qquad (5)$$

Let $\mu_{\pi,t,l}(.)$ be the state distribution at stage $l$ of round $t$ under policy $\pi$. For $x = (s,a_0,n_1,a_1,\ldots,n_l)$, we can write

$$\mu_{\pi,t,l}(x) = \pi(a_0|s)\,\mathbb{I}_{\{g_t(s,a_0)=n_1\}}\cdots\pi(a_{l-1}|n_{l-1})\mathbb{I}_{\{g_t(n_{l-1},a_{l-1})=n_l\}}\,.$$

Introduce the notation

$$\pi(a_{0\ldots(l-1)}|x) = \pi(a_0|s)\ldots\pi(a_{l-1}|n_{l-1})\,,\quad \mathbb{I}_{\{g_t(n_{1\ldots l}|x)\}} = \mathbb{I}_{\{g_t(s,a_0)=n_1\}}\cdots\mathbb{I}_{\{g_t(n_{l-1},a_{l-1})=n_l\}}\,.$$

We have that

$$\mathbb{E}\left[\sum_{t=1}^{T}\sum_{l=1}^{L}(Q_t(x_{t,l}^\pi,\pi_t) - Q_t(x_{t,l}^\pi,\pi))\right] = \sum_{t=1}^{T}\sum_{l=1}^{L}\sum_{x\in\mathcal{X}_l}\mu_{\pi,t,l}(x)(Q_t(x,\pi_t) - Q_t(x,\pi))$$

$$= \sum_{l=1}^{L}\sum_{x\in\mathcal{X}_l}\sum_{t=1}^{T}\mu_{\pi,t,l}(x)(Q_t(x,\pi_t) - Q_t(x,\pi))$$

$$= \sum_{l=1}^{L}\sum_{x\in\mathcal{X}_l}\sum_{t=1}^{T}\pi(a_{0\ldots(l-1)}|x)\mathbb{I}_{\{g_t(n_{1\ldots l}|x)\}}(Q_t(x,\pi_t) - Q_t(x,\pi))$$

$$= \sum_{l=1}^{L}\sum_{x\in\mathcal{X}_l}\pi(a_{0\ldots(l-1)}|x)\sum_{t=1}^{T}\mathbb{I}_{\{g_t(n_{1\ldots l}|x)\}}(Q_t(x,\pi_t) - Q_t(x,\pi))\,,$$

where the last step follows from the fact that $\pi(a_{0\ldots(l-1)}|x)$ does not depend on time. Thus, we can write

$$\mathbb{E}\left[\sum_{t=1}^{T}\sum_{l=1}^{L}(Q_t(x_{t,l}^\pi,\pi_t) - Q_t(x_{t,l}^\pi,\pi))\right] \le \sqrt{T\log\frac{|\mathcal{A}|}{2}}\sum_{l=1}^{L}\sum_{x\in\mathcal{X}_l}\pi(a_{0\ldots(l-1)}|x)$$

$$\le L\,|\mathcal{G}|\,\sqrt{T\log\frac{|\mathcal{A}|}{2}}\,, \qquad (6)$$

where the first step follows from the regret bound for the EWA algorithm [16] and the second step follows from Lemma 13.

Finally, by an application of Azuma's inequality, we obtain that

$$\sum_{l=1}^{L}\left(\mathbb{E}\left[Q_t(x_{t,l}^\pi,\pi_t)\,|\,x_{t,l-1}^\pi\right] - Q_t(x_{t,l}^\pi,\pi_t)\right) \le L\sqrt{8T\log(2T)} + L\,. \qquad (7)$$

From (5),(6),(7), we obtain the desired result:

$$\mathbb{E}\left[\sum_{t=1}^{T}\sum_{l=1}^{L}(Q_t(s,\pi_t)/L - c_t(x_{t,l}^\pi,\pi))\right] \le L\,|\mathcal{G}|\,\sqrt{T\log\frac{|\mathcal{A}|}{2}} + L\sqrt{8T\log(2T)} + L\,.$$

$$\square$$

## B.2  Bounding $\mathbb{E}\left[B_T\right]$

**Lemma 15.** *We have that* $\mathbb{E}\left[B_T\right] \le L\sqrt{8T\log(2T)} + L$.

*Proof.* We have that $Q_t(x_{t,l}^{\pi_t}, \pi_t) = \mathbb{E}\left[c_t(x_{t,l}^{\pi_t}, \pi_t) + Q_t(x_{t,l+1}^{\pi_t}, \pi_t) \,\middle|\, x_{t,l}^{\pi_t}\right]$. Thus,

$$
\begin{aligned}
Q_t(x_{t,l}^{\pi_t}, \pi_t) - \mathbb{E}\left[Q_t(x_{t,l}^{\pi_t}, \pi_t)|x_{t,l-1}^{\pi_t}\right] &= \mathbb{E}\left[c_t(x_{t,l}^{\pi_t}, \pi_t) \,\middle|\, x_{t,l}^{\pi_t}\right] \\
&+ \mathbb{E}\left[Q_t(x_{t,l+1}^{\pi_t}, \pi_t)|x_{t,l}^{\pi_t}\right] - \mathbb{E}\left[Q_t(x_{t,l}^{\pi_t}, \pi_t)|x_{t,l-1}^{\pi_t}\right] .
\end{aligned}
$$

For episode $t$,

$$
\begin{aligned}
\sum_{l=1}^{L} \left(Q_t(x_{t,l}^{\pi_t}, \pi_t) - \mathbb{E}\left[Q_t(x_{t,l}^{\pi_t}, \pi_t)|x_{t,l-1}^{\pi_t}\right]\right) &= \sum_{l=1}^{L} \mathbb{E}\left[c_t(x_{t,l}^{\pi_t}, \pi_t) \,\middle|\, x_{t,l}^{\pi_t}\right] \\
&+ \sum_{l=1}^{L} \left(\mathbb{E}\left[Q_t(x_{t,l+1}^{\pi_t}, \pi_t)|x_{t,l}^{\pi_t}\right] - \mathbb{E}\left[Q_t(x_{t,l}^{\pi_t}, \pi_t)|x_{t,l-1}^{\pi_t}\right]\right) \\
&= -Q_t(s, \pi_t) + \sum_{l=1}^{L} \mathbb{E}\left[c_t(x_{t,l}^{\pi_t}, \pi_t) \,\middle|\, x_{t,l}^{\pi_t}\right] .
\end{aligned}
$$

Thus,

$$
\sum_{t=1}^{T}\sum_{l=1}^{L} \left(\mathbb{E}\left[c_t(x_{t,l}^{\pi_t}, \pi_t) \,\middle|\, x_{t,l}^{\pi_t}\right] - Q_t(s, \pi_t)/L\right) = \sum_{t=1}^{T}\sum_{l=1}^{L} \left(Q_t(x_{t,l}^{\pi_t}, \pi_t) - \mathbb{E}\left[Q_t(x_{t,l}^{\pi_t}, \pi_t)|x_{t,l-1}^{\pi_t}\right]\right) .
$$

Thus, by an application of Azuma's inequality, we obtain that

$$
\mathbb{E}\left[\sum_{t=1}^{T}\sum_{l=1}^{L} (c_t(x_{t,l}^{\pi_t}, \pi_t) - Q_t(s, \pi_t)/L)\right] \leq L\sqrt{8T\log(2T)} + L .
$$

$\square$

*Proof of Theorem 7.* The result is obvious by Lemmas 14 and 15. $\square$

## Footnotes

[5]Notice that $\mathcal{F}_T$ contains only policies, which are independent of the state variables.