[Reviews · NeurIPS 2013]

Submitted by Assigned_Reviewer_4

Summary:
On its first part the paper deals with online learning of Markov Decision Processes. Given an MDP and a set of policies (=distributions over actions) how should one Summary:
The first part of the paper addreses the problem of Online Learning in Markov Decision Process.
In this problem at each iteration:
1) adversary chooses a loss function and transition kernel (probabilistic function that chooses the next state based on the currenone and the selected action)
2) the leaner knows the current state x_t and chooses an action a_t.
3) the leaner sees the adversary choices and suffers the loss
4) the next state is sampled.
The goal of the learner is to minimize the regret w.r.t. PI - a set of action selction policies
The authors propose an algorithm which is a variant of the Shrinking Dartboard algorithm and show a bound of O((Tlog|PI|)^{1/2}).

The second part of the paper addreses the Adersarial Online Shortest Path Problem. Here at each iteration the learner is presented with a graph with source, destinations and all other nodes in layers structure.
The Learner chooses a path from source to dest. and then the weights of the edges is revealed and the loss of the learner is defined as the sum of weights along the chosen path.
The goal of the learner is to minimize the regret w.r.t. a set of path choosing policies.
The main contribution on this part is by showing that if both the graph and weights are chosen by an adversary at each stage this problem is as hard as the online agnostic parity problem (which is assumed to be a hard problem).

Quality & Clarity

The results presented in the paper seem new and interesting to me.
However, 2 points bothered me during the reading:
1) It would be helpful if the explanation of the difference between the online MDP problem and standard online problems (lines 200-204) will appear much earlier and would be a bit longer than 2 sentences.
2) The connection between the 2 parts of the paper is not obvious to the learner. The authors state that Online Shortest Path is a special case of MDP. I assume that this is correct but expect that such claim will be accompanied by a short explanation (States are:..., Actions are:... Loss function:...)
(I haven't read the Supplementary material)

Originaluty & Significance
I'm not familiar with previous results on this problem.
The hardness proof might be useful for other works in the area


Comments:
1)
In Remark 3 the authors claim that a bound for determintic policies won't apply a bound w.r.t. the optimal policy. I'm not sure that I understand this claim: as far as I know a regret of an algorithm is always measured w.r.t. a set of competing policies. Hence, a bound on the regret of every deterministic policy will imply an optimal regret w.r.t. any deterministic policy. I undrstand that there might be some stochastic policy which is better but this policy is not part of the competing group of policies.
2)
In the hardness results the authors mention several times "noisy parity" however, I didn't find where the noise is mentioned. (Did you mean that the labeling y is sometime flipped? Is it matters how is is flipped?)


Typos:
Line 64 defines policy as mapping from actions intead of a mapping from states (as in line 135)
Figure1:OMDP algo: W_0 is undefined, w_{pi_0},-1 is undefined.
Line 178: This notion of regret w.r.t. specific policy wasn't defined previously. (and appears a lot in the paper)
Line 367: A denotes an algorithm while earlier it was used for the set of possible actions.
Line 412: Did you define earlier N as the set of all nodes?
Line 412 at the end: shouldn't the l in Q_{pi, g l} be changed to c?
Summary: A good paper, however the presentation of the results should be improved.

Submitted by Assigned_Reviewer_5

The paper is about adversarial settings of MDPs.

While such problems as stochastic online prediction and bandits
have their adversarial formulations, the problem of learning
MDPs thus far had only limited adversarial formulations: the transitions
were assumed stochastic and the only rewards were considered adversarial.
The present paper looks into what can be done if the transitions are adversarial as well.

It presents some positive results and a negative (hardness) result.

The main positive result concerns a setting where at each step the transition
probability kernel is selected by an adversary, and is entirely revealed after
the agent has made an action. The performance is compared to the imaginary
performance of a set of experts that experience the same sequence of transition kernels.
Under some additional assumptions, notably, a uniform (over all policies and all transition
kernels) mixing assumption, a sqrt(T) regret bound is obtained for an online-learning-style algorithm
presented.

The most important feature in this setting is that the transition kernel is revealed
after each action. This makes the the expected loss of each expert available to the learner.
This feature allows the authors to reduce the problem to the online learning problem.
Most of the "learning" part of the problem is thus removed: what has to be learned
concerns the loss of the experts over time, rather than something about the "MDP."
In particular, the choice of the resulting policy is independent of the observed states.

I think this makes the setting rather limited. However, the fact that the setting
for the positive result is limited is justified by the negative result presented.

In addition, I think that labelling this problem "adversarial" is somewhat misleading,
since the sequence of transition kernels is fixed and is the same for all experts.
Had the adversary been really playing against each of the experts, he could have chosen
a different sequence for each of them, changing the performance/losses completely.
However, this general remark applies to all problems called "adversarial" in the literature,
and therefore I do not blame the authors for using this terminology.


I think this negative (hardness) result is the most interesting part of the paper. It shows
that learning noisy parities - a notoriously hard problem used in cryptography -
can be reduced to an adversarial version of the shortest path problem. The latter
is a special case of learning MDPs with adversarial transitions.
The authors point out that another hardness result had previously been claimed
but later retracted.


The comparison class of experts in this adversarial shortest path problem is that of all
deterministic policies. For general classes of experts the authors also provide a regret bound
that scales linearly with the number experts (so it is useful if the class of experts is much
smaller than that of all deterministic policies).

To summarize, the paper provides some positive and a negative results concerning
learning in MDPs with adversarially chosen transitions. While the positive results appear
somewhat limited, altogether the paper gives a good insight in what is possible and what is not
possible in this problem. The hardness result appears especially interesting.
Summary: To summarize, the paper provides some positive and a negative results concerning
learning in MDPs with adversarially chosen transitions. While the positive results appear
somewhat limited, altogether the paper gives a good insight in what is possible and what is not
possible in this problem. The hardness result appears especially interesting.

Submitted by Assigned_Reviewer_6

The authors consider a general class of MDPs where loss functions and
state dynamics are chosen by an adversary. The derive an algorithm
with a guarantee of sublinear regret with respect to a bounded set of
competing policies. Additionally, an algorithm with sublinear regret
is derived for an episodic version of the shortest path
problem. Lastly, a hardnress result is presented for the design of
efficient sublinear regret algorithms for the fully adversarial
version of the online shortest path problem.

Most of the paper consists of theorem statements and a discussion of
their implications. Proofs are left to the supplementary section,
although proof sketches are provided.

Thm 1 provides a sublinear regret bound that has a term which is
scaled by the inverse of the bound on the mixing time plus an additive
term related to the size of the reference policy set.

The discussion in section 4 regarding Thm 5 is interesting. Given that
the SPP is a special case of the online MDP problem the implication
that a computationally efficient algorithm exists, although this
appears to be an informated conjecture rather than a proof.

The paper as written is clear and interesting. Unfortunately, it
appears to be incomplete. It ends abruptly at the statement of theorem
7 with no follow up discussion. I am assuming a formatting issue. If
not for this, my rating would be higher.

minor comments:

Thm 1 - I assume that R_T is the regret after T rounds, but this is not defined until later (and only implicitly).

Summary: REPLACE THIS WITH YOUR ANSWER
Author Feedback

Author rebuttal: We thank all reviewers for their helpful feedback.

Assigned_Reviewer_4:

We will explain in more details the differences between online MDP and standard online problems and also make the connection between the 2 parts of the paper more clear. Thanks also for spotting the typos. We will correct them.

In Remark 3, by optimal policy, we meant the optimal policy in the class of stationary policies. We wanted to emphasize that a low regret wrt deterministic policies does not imply a low regret wrt stationary policies. We will clarify this distinction.

Noisy Parity: Here, we mean that the labels may be arbitrary (agnostic) and the goal is to predict competitively with respect to the "best" parity function. We appreciate the point that the current write-up may be a bit confusing to NIPS-readers not familiar with computational learning theory, and will improve the writing in the next version.

Assigned_Reviewer_5:

It is true that the choice of policy is independent of the observed state, but it is not clear that this is really a limitation. At the end the algorithm outputs a policy, which takes the state into account. The regret suffered by the algorithm is defined in terms of the actual sequence of states encountered. We acknowledge that observing transition kernels can be a strong assumption in some problems. Estimating expected losses without such an assumption seems very challenging.

Assigned_Reviewer_6:

We will save space elsewhere to be able to add a short discussion after Theorem 7. We think this is relatively easy to fix.